# Potent GST Ketosteroid Isomerase Activity Relevant to Ecdysteroidogenesis in the Malaria Vector *Anopheles gambiae*

**DOI:** 10.3390/biom13060976

**Published:** 2023-06-11

**Authors:** Yaman Musdal, Aram Ismail, Birgitta Sjödin, Bengt Mannervik

**Affiliations:** 1Department of Biochemistry and Biophysics, Stockholm University, SE-10691 Stockholm, Sweden; ymusdal@gmail.com (Y.M.); aram.ismail@dbb.su.se (A.I.); birgitta.sjodin@dbb.su.se (B.S.); 2Department of Pediatric Genetics, Faculty of Medicine, Hacettepe University, 06230 Ankara, Turkey; 3Department of Chemistry, Scripps Research, La Jolla, CA 92037, USA

**Keywords:** Nobo, *Anopheles gambiae* GSTE8, malaria, ketosteroids, ecdysteroidogenesis

## Abstract

Nobo is a glutathione transferase (GST) crucially contributing to ecdysteroid biosynthesis in insects of the orders *Diptera* and *Lepidoptera*. Ecdysone is a vital steroid hormone in insects, which governs larval molting and metamorphosis, and the suppression of its synthesis has potential as a novel approach to insect growth regulation and combatting vectors of disease. In general, GSTs catalyze detoxication, whereas the specific function of Nobo in ecdysteroidogenesis is unknown. We report that Nobo from the malaria-spreading mosquito *Anopheles gambiae* is a highly efficient ketosteroid isomerase catalyzing double-bond isomerization in the steroids 5-androsten-3,17-dione and 5-pregnen-3,20-dione. These mammalian ketosteroids are unknown in mosquitoes, but the discovered prominent catalytic activity of these compounds suggests that the unknown Nobo substrate in insects has a ketosteroid functionality. Aminoacid residue Asp111 in Nobo is essential for activity with the steroids, but not for conventional GST substrates. Further characterization of Nobo may guide the development of new insecticides to prevent malaria.

## 1. Introduction

The glutathione transferase (GST) enzyme Nobo is essential to the biosynthesis of the molting hormone ecdysone in the mosquito *Anopheles gambiae*. The mosquito is a prominent vector of *Plasmodium falciparum*, the parasite causing malaria, and several hundred thousand people die from malaria every year. The disease is spread by bites of parasite-carrying mosquitoes, thereby threatening billions of people in >100 countries world-wide. Transmission of the infectious agent is generally prevented by control of the vector by insecticides, mosquito nets, and the elimination of breeding sites [1]. However, the spread of insecticide resistance has led to general agreement that malaria control is in urgent need of novel agents [2].

The proliferation of mosquitoes is critically dependent on the steroid hormone ecdysone, and the GST enzyme called Nobo (encoded by the gene *Noppera-bo*) was first discovered in the fruit fly *Drosophila melanogaster* to be essential for ecdysteroid biosynthesis [3,4]. Nobo is expressed abundantly in the major ecdysone-producing tissues, primarily in the prothoracic gland, but also in the testis and the ovary [4,5]. Nobo knockout *D. melanogaster* presents with embryonic lethality and a naked cuticle structure—phenotypes typical for mutants showing embryonic ecdysteroid deficiency. Furthermore, Nobo knockdown larvae displayed lowered titers of the ultimate hormone 20-hydroxyecdysone. A corresponding Nobo GST has also been discovered in the mosquitoes *Aedes aegypti* and *Culex quinquefasciatus* [4], which, like *An. gambiae*, are important vectors of infectious diseases. 

A flavonoid inhibitor of Nobo from *Ae. aegypti* was recently demonstrated to have larvicidal activity, thus supporting the notion that Nobo inhibitors could find use as novel insecticides [6]. Notably, Nobo is present only in dipteran and lepidopteran insects and not in other life forms including humans [4]. From this perspective, the enzyme is a perfect new target for new preventative and selective insecticides, avoiding collateral toxicity to honeybees and other valuable biological species. In spite of its demonstrated importance for ecdysteroid biosynthesis, the biochemical function of Nobo is unknown. We have initiated investigations of Nobo from the mosquito *An. gambiae*, the major vector of the pathogen *P. falciparum*. A better understanding of the enzymatic properties can form the basis of the discovery and design of potent and selective Nobo inhibitors for use in combating the malaria vector.

## 2. Materials and Methods

### 2.1. Materials

The steroids 5-androsten-3,17-dione and 5-pregnen-3,20-dione were purchased from Steraloids Inc. (Newport, RI, USA). All other chemicals were obtained from Sigma-Aldrich (St. Louis, MO, USA).

### 2.2. Methods

#### 2.2.1. Extracting Plasmid DNA from Filters

DNA coding for Nobo protein XP_319963.1 from *An. gambiae* str. PEST (also known as the enzyme GSTE8) was synthesized by ATUM (Newark, CA, USA). The synthetic gene encodes the same protein sequence as the natural gene AGAP009190-PA located on chromosome 3R. However, alternative codons promoting high-level expression in *Escherichia coli* were used in the synthesis according to ATUM’s proprietary algorithms. The gene was ligated into the pD444-SR expression plasmid and delivered on a GF/C glass microfiber filter. Site-specific single-point mutants of Nobo were similarly produced by ATUM via total chemical synthesis of the corresponding genes. The DNA was extracted with 100 μL 10 mM Tris-HCl, pH 7.5, and centrifuged for 1 min at 10,000× g, yielding 90 μL containing the extracted DNA at a concentration of 20 ng/μL.

#### 2.2.2. Transformation of Bacteria

*E. coli* BL21 (DE3) cells were transformed with 2 μL (40 ng) of the extracted DNA. The cells were kept on ice for 30 min before being heat shocked in a 42 °C water bath for 45 s and then put on ice for 2 min. LB broth (250 μL) was added and the bacteria were grown at 37 °C with shaking for 45 min. The transformed cells were plated on agar with ampicillin (50 μg/mL) and incubated overnight at 37 °C.

#### 2.2.3. Expression and Purification of Recombinant GSTs

An overnight culture was prepared from a colony of transformed bacteria suspended in 50 mL LB medium containing ampicillin (50 μg/mL) and incubated overnight at 37 °C with 200 rpm shaking. A flask of 500 mL expression medium 2 **×** TY (8 g bacto-tryptone, 5 g yeast extract, 2.5 g NaCl, and 25 mg ampicillin) was inoculated with 5 mL overnight culture and grown at 37 °C, 200 rpm. When OD_600_ reached 0.4, enzyme expression was induced with 0.2 mM isopropyl β-D-1-thiogalactopyranoside and the bacteria were further grown for 3 h at 37 °C. The culture was then centrifuged for 7 min at 7000× *g*. The bacterial pellet was mixed with 10 mL lysis buffer (20 mM sodium phosphate, 20 mM imidazole, 0.5 M NaCl pH 7.4, 0.2 mg/mL lysozyme, and one complete mini tablet of protease inhibitor cocktail) and incubated for 1 h. Finally, the cells were disrupted by sonication and the lysate was centrifuged for 30 min at 27,000× *g*.

The expressed GSTs contained an N-terminal hexahistidine tag, enabling their purification by immobilized metal affinity chromatography (IMAC) using a Ni-IMAC column (GE Healthcare, Uppsala, Sweden). The supernatant fraction from the centrifugation was loaded onto the column equilibrated with binding buffer (20 mM sodium phosphate, 20 mM imidazole, 0.5 M NaCl, pH 7.4), and the column was rinsed with the binding buffer to remove unbound material. An elution buffer (20 mM sodium phosphate, 500 mM NaCl, 250 mM imidazole, pH 7.4) was used to release GST from the column. The eluted GST was dialyzed two times against 10 mM Tris-HCl, pH 7.8, 1 mM EDTA, and 0.2 mM tris(2-carboxyethyl)phosphine. SDS-PAGE (sodium dodecyl sulfate-polyacrylamide gel electrophoresis) verified the purity of the dialyzed enzyme.

#### 2.2.4. Kinetic Experiments

Enzyme activities were determined with a selection of standard GST substrates [7]. Measurements were performed spectrophotometrically at 30 °C as described in detail previously, including the wavelengths and extinction coefficients used to determine reaction rates [8,9]. However, the assay systems were supplemented with 0.1% (*w/v*) bovine serum albumin, which gave more reproducible measurements. It should be noted that due to limitations of solubility, 5-AD was tested at 0.10 mM, whereas 5-PD was limited to 0.010 mM in the determination of specific activities. The inhibition experiments were performed in 96-well plates at pH 7.5 in 0.1 M sodium phosphate buffer with 1.0 mM DCNB, 5.0 mM glutathione, and 17β-estradiol dissolved in ethanol (5% final concentration in the assay system). Kinetic data were evaluated by nonlinear regression analysis using GraphPad Prism 9.

## 3. Results

A gene encoding the 217 amino acid sequence of Nobo from *An. gambiae* (also named AgaGSTE8) was synthesized and ligated into a plasmid for expression in *Escherichia coli*. The enzyme purified by Ni-IMAC was estimated as >95% homogeneous by SDS-PAGE. Gel filtration verified the purity of Nobo and showed the molecular mass of approximately 50 kDa expected for a dimeric GST protein.

The catalytic activity of the purified enzyme was assayed with standard GST substrates [10]: two aryl halides, 1-chloro-2,4-dinitribenzene (CDNB) and 1,2-dichloro-4- nitrobenzene (DCNB); cumene hydroperoxide (CuOOH); and two isothiocyanates, phenethyl isothiocyanate (PEITC) and allyl isothiocyanate (AITC). The specific activities with these conventional substrates varied between 0.4 and 25 µmol/min per mg protein in the range common to many GST enzymes (Table 1).

Remarkably, the double-bond isomerase activity with 5-androsten-3,17-dione (5-AD) was 245 µmol/min per mg, significantly higher than the specific activity with any other substrate (Table 1). Similar efficient ketosteroid isomerase catalysis was demonstrated with the alternative steroid substrate 5-pregnen-3,20-dione (5-PD) measured at a 10-fold lower concentration (Table 1). Glutathione was an essential cofactor, although not consumed in the steroid reactions, as earlier demonstrated for mammalian GSTs [11]. Previous studies have identified high double-bond isomerase activity in human [12] and equine GST A3-3 [8], indicating the involvement of GST A3-3 in steroid hormone biosynthesis [13]. A steady-state kinetic analysis of Nobo with 5-AD as the varied substrate determined kcat as 291 ± 6 s^−1^ and kcat/Km as (6.9 ± 0.3) **×** 10^6^ M^−1^s^−1^.

Modeling of the *An. gambiae* Nobo protein was performed using the homology with the crystal structures of Nobo from *Ae. aegypti* (PDB ID: 7EBW) as well as from *D. melanogaster* (PDB ID: 6KEP). The *An. gambiae* models were highly similar even though the template sequences between themselves were only 41% identical. The similarities were particularly prominent in the regions of protein secondary structure and in the location of the cofactor glutathione. The second ligand, desmethylglycitein (4′,6,7-trihydroxyisoflavone) in 7EBW and 17β-estradiol in 6KEP, is a polycyclic molecule occupying essentially the same position in the protein structure, juxtaposed to the bound glutathione. Figure 1 shows the *An. gambiae* model derived from 6KEP with the two ligands 17β-estradiol and glutathione superpositioned from the template structure.

Examination of the *An. gambiae* Nobo model suggested four residues of possible importance for catalytic activity: Ser9, His40, Asp111, and Arg116. The following single-point mutants of the enzyme were therefore synthesized: D111N, S9A, H40N, and R116A. In D111N, the acidic Asp is replaced with the isosteric but uncharged Asn, and in H40N His is similarly changed to Asn, which cannot act as an acid–base catalyst. The substitution S9A removes the hydroxyl group of Ser, supposedly forming a hydrogen bond to the sulfur of glutathione bound to the active site [6]. In R116A, removal of the positively charged guanidinium group of Arg prevents the formation of an ionic bond with a carboxyl group of glutathione. All mutant proteins expressed appeared properly folded and the purified mutant enzymes were tested with the alternative substrates (Table 1).

Notably, the D111N mutation did not markedly change the specific activities with the conventional GST substrates. However, the ketosteroid isomerase activity was dramatically diminished by two orders of magnitude with both 5-AD and 5-PD. The mutation S9A decreased the activity with most substrates by 10- to 20-fold, but 100-fold with CuOOH (Table 1). The mutations H40N and R116A clearly reduced the activities with the ketosteroids, aryl halides, and with CuOOH, but had only a small effect on the isothiocyanate reactions.

In previous studies on Nobo from *D. melanogaster* and *Ae. aegypti*, 17β-estradiol was found to be an inhibitor of the enzyme, demonstrating an affinity for steroids [6,14]. The Nobo from *An. gambiae* was likewise inhibited by estradiol with an IC_50_ value 0.17 ± 0.03 µM (Figure 2). Notably, the D111N mutant was significantly less sensitive to the inhibitor, showing an IC_50_ value of 15.5 ± 6.0 µM.

### Nobo Is an Efficient Ketosteroid Isomerase

The bacterial ketosteroid isomerase is among the most efficient of all enzymes known, with a specific activity of the *Pseudomonas testosterone* enzyme reported as 45,300 µmol/min per mg and the catalytic efficiency kcat/Km 1.58 × 10^8^ M^−1^s^−1^ [15], approaching the diffusion limit of enzyme reactions [16]. The corresponding values for *An. gambiae* Nobo, 245 µmol/min per mg and 6.9 **×** 10^6^ M^−1^s^−1^, are significantly lower, but the mosquito enzyme still qualifies for the upper echelon of highly active enzymes [17]. The requisite in Nobo for the obligatory glutathione in the active site may limit the catalytic turnover as compared with the bacterial ketosteroid isomerase, which is lacking a cofactor. The steroids 5-AD and 5-PD, shown to be excellent substrates of Nobo, are not intermediates in insect ecdysteroidogenesis, but by mechanistic analogy their efficient isomerization suggests that the enzyme catalyzes a similar reaction in the “black box” of ecdysone biosynthesis [18]. The finding that glutathione is required for ecdysteroidogenesis [19] lends further support to the notion that the glutathione-dependent Nobo fulfills a catalytic role in the biological context.

The modeling of *An. gambiae* Nobo based on the crystal structures of the homologous enzymes from *Ae. aegypti* and *D. melanogaster* demonstrates that 5-AD (or 5-PD) can fit in an active site pocket apparently reaching the carboxyl group of Asp111 (Figure 1). A corresponding carboxylate (Asp113 in *D. melanogaster* and Glu113 in *Ae. aegypti*) is considered a salient and conserved signature feature of all 21 identified Nobo GSTs in *Diptera* and *Lepidoptera* [14]. In the *D. melanogaster* Nobo, binding of 17β-estradiol occurs via its 3-hydroxyl group to Asp113 in the conserved site [14], but binding affinity is lost upon the mutation of Asp113 into Ala, which eliminates the carboxylate responsible for hydrogen bonding. Our demonstration that the D111N mutation in *An. gambiae* Nobo causes a 100-fold loss of affinity for 17β-estradiol as indicated by the increased IC50 indicates a corresponding interaction with the inhibitor (Figure 2). We propose that Asp111 makes a similar bond to the 3-keto group of the ketosteroid substrate, thereby promoting the formation of a dienolate intermediate (Figure 3). The thiolate group of glutathione bound to Nobo is stabilized by hydrogen bonding to Ser9, like in other GSTs featuring Ser in the active site [20]. The activated thiolate close to the steroid in the active site could serve as a base and abstract a proton from the C4 position, as previously shown in the mammalian GST A3-3 [21].

The reaction is then completed by addition of a proton to C6 in the steroid. The substantial loss of ketosteroid isomerase activity effected by mutations D111N and S9A in *An. gambiae* Nobo are consistent with these proposed functions of the active-site residues (Table 1). The modeled structure indicates that His40 could contribute to the binding of glutathione by interacting with the C-terminal carboxylate group of the tripeptide. Arg116 is located at the other side of the same carboxylate and near the orifice of the proposed binding site of the steroid substrate, and like His40 is distant from the residues directly involved in catalysis. Thus, the H40N and R116A mutations remote from the site of chemical transitions in the steroid substrate only modestly diminish the catalytic activity. 

Our proposed mechanism for Nobo in the steroid double-bond isomerization is similar to the bacterial ketosteroid isomerase reaction by invoking a carboxylate, which polarizes C=O in position C3 of the steroid substrate [22]. Like in the mammalian GSTs with steroid isomerase activity [23], the glutathione thiolate in Nobo serves as the base attacking the C4 hydrogen. In other respects, the enzymes differ in functional groups even though they catalyze the same isomerization reaction. The primary structures of the Nobo proteins and the mammalian GSTs segregate into widely separated evolutionary clades [14], and neither of them shows any structural relationship to the bacterial ketosteroid isomerase.

## 4. Discussion

Every year more than half a million people die of malaria and other diseases whose pathogenic agents are spread by mosquito bites. The most severe form of human malaria is caused by *P. falciparum*, which is transmitted by mosquitoes of the *Anopheles* genus, and the disease is endemic in 76 countries. The transmission of pathogens is largely prevented by controlling vector populations by spraying insecticides, the use of impregnated mosquito nets, and the elimination of breeding sites. Malaria control relies heavily on insecticides via indoor residual spraying and long-lasting insecticidal bed nets targeting the principal African vector *An. gambiae*. However, both resistance and climate change threaten to reverse the progress made by insecticidal mosquito control in recent years. Disappointingly, the WHO reports that resistance of the *Anopheles* vectors has emerged against the four main insecticide classes used for mosquito control, i.e., pyrethroids, organochlorines, carbamates, and organophosphates [1]. Malaria vectors are able to resist the actions of insecticides due to various resistance mechanisms, including target site mutations, cuticular modification, and metabolic inactivation. The latter occurs by the selection of mosquitoes with upregulated or more efficient endogenous insecticide detoxifying enzymes. Obviously, novel means to combat the *Anopheles* vectors transmitting the deadly pathogen are in urgent demand [24]. 

GSTs are ubiquitous and abundant proteins performing a wide range of enzymatic and non-enzymatic functions [25]. In insects, GSTs form an important detoxication system comprising numerous enzymes involved in the metabolism of a wide range of foreign and endogenous compounds [26]. GSTs play an essential role in insect herbivory through the detoxification of deterrent and toxic plant allelochemicals. Research on GSTs in insects was initially motivated by their plausible involvement in insecticide resistance [27], since elevated GST activity had been detected in strains of insects that are resistant to organophosphates and organochlorines [28]. Insect GSTs have been divided into six classes of cytosolic enzymes: Delta, Epsilon, Omega, Sigma, Theta, and Zeta [29]. In addition, membrane-bound insect GSTs exist, similar to mammalian MAPEG proteins [30]. Members of the Delta- and Epsilon-class GSTs have been associated with the detoxication of various chemicals, including DDT, an insecticide used in indoor residual spraying. The *An. gambiae* GST Nobo in the present investigation (also called GSTE8 [2]) belongs to the Epsilon class and its orthologs in *D. melanogaster* and *Ae. aegypti* are designated GSTE14 and GSTE8, respectively. It is noteworthy that neither Delta nor Epsilon GSTs are present in humans and other mammals or in plants. 

Ecdysone is a steroid that serves as a master regulator in the control of development, involving molting, metamorphosis, and diapause, in ecdysozoan animals, which encompass insects and nematodes [18]. Following its synthesis, the ecdysone molecule is distributed to various tissues, where it is hydroxylated to form 20-hydroxyecdysone. Transcriptional gene regulation is effected by the binding of 20-hydroxyecdysone to a cognate nuclear receptor in target tissues. The biosynthesis of ecdysone originates in dietary cholesterol, and the accepted biosynthetic pathway is initiated by the 7,8-dehydrogenation of cholesterol and is terminated by a series of reactions catalyzed by different cytochrome P450 (Halloween) enzymes, leading to ecdysone and finally 20-hydroxyecdysone. In between, a number of chemical transformations take place, which remain partly unknown (black box in Figure 4).

Following the dehydrogenation of cholesterol leading to 7-dehydrocholesterol (7dC), it has been hypothesized that reactions in the black box involve oxidation of the C3 hydroxyl group in 7dC to yield 3-oxo-7dC (cholesta-5,7-diene-3-one), which may undergo double-bond isomerization to cholesta-4,7-diene-3-one [31]. Both 3-oxo-7dC isomers are unstable, but it has been demonstrated that photosensitive, ketone-blocked derivatives release 3-oxo-7dC upon irradiation with UV light of a wavelength that is innocuous to the insect. Administered in vivo by this photolytic approach, 3-oxo-7dC is efficiently incorporated into ecdysteroids by both *D. melanogaster* and the tobacco hornworm moth *Manduca sexta* [31]. The three reactions leading from 5β-ketodiol to ecdysone are catalyzed by known cytochrome P450 enzymes present in the prothoracic gland. Ecdysone is released into the hemolymph and oxidized to 20-hydroxyecdysterone in peripheral tissues [32].

In 2014, two research groups identified the gene product of *Noppera-bo* as GSTE14 in *D. melanogaster* [3,4]. Like Halloween cytochrome P450 gene knockouts, null alleles of *Noppera-bo* result in embryonic lethality, embryonic cuticle abnormalities, and decreased ecdysteroid concentrations. Curiously, administration of either cholesterol or 20-hydroxyecdysterone rescues the phenotype, leaving the role of the enzyme Nobo (GSTE14) and the effect of knockdown of the gene unexplained. Niwa and coworkers have identified orthologous GSTs of the Epsilon class in numerous insects, but the presence of these orthologs is limited to the orders *Diptera* and *Lepidoptera* [14].

In addition to the well-established roles of ecdysteroids in metamorphosis and molting, recent investigations further demonstrate the crucial importance of ecdysone signaling for the insect proliferation [33]. For mosquitoes in particular, both oogenesis and larval development are critically dependent on ecdysone in both *An. gambiae* and *Ae. aegypti* [34]. Genetic modification of *An. gambiae* has demonstrated that disruption of the ecdysone pathway is associated with embryonic lethality midway through development [35]. Furthermore, a recent study of mosquito larvae demonstrated that larvicidal activity correlated with the inhibition of Nobo [6]. Since ecdysteroids are essential to molting and proliferation in insects, their biosynthesis is therefore a possible target for insect growth regulation and pest control. 

The spread of insecticide resistance in *Anopheles* species has prompted the search for alternative sustainable approaches to controlling malaria. Pyrethroid resistance is, according to the WHO, currently the greatest biological threat to malaria control. Although new bed nets containing pyrethroid synergists or additional insecticides are slowly being introduced to tackle the loss of insecticidal activity of existing pyrethroid-only nets against increasingly resistant mosquitoes [36], it is clear that malaria control is in dire need of novel agents [24].

The structure of Nobo from *D. melanogaster* has been solved [9,14], and the enzyme was found to interact with steroid molecules structurally similar to the insect hormone ecdysone and its precursor cholesterol. We discovered that the *D. melanogaster* enzyme catalyzes a ketosteroid double-bond isomerization of 5-androsten-3,17-dione [9], but the catalytic efficiency was 20-fold lower than that of Nobo from *An. gambiae* reported here. 

The GST Nobo enzyme is fundamental to the proliferation of dipteran and lepidopteran insects, being crucial for metamorphosis and oogenesis [37], and inhibitors of the enzyme can be expected to have similar lethal effects as demonstrated in homozygous Nobo null and Asp113Ala mutants in *D. melanogaster* [14]. These deleterious effects on ecdysteroidogenesis are noted by Nobo knockout or inhibition in a range of distantly related insect species. For example, similar lethal phenotypes are observed in *Bombyx mori* Nobo null insects [5]. In mosquitoes, inhibitors of the Nobo ortholog from the yellow-fever-transmitting mosquito *Ae. aegypti* are larvicidal to this insect and suppress the expression of the ecdysone-inducible gene E74B [6].

Globally, the number of malaria cases has stayed in excess of 200 million every year since 2000, but the tally of deaths has declined from 738,000 in the year 2000 to 409,000 in 2019. However, the number of fatalities has reportedly risen to 619,000 in 2021 according to the WHO. In spite of some reduced mortality in this period, the rate of improvement has stalled in many countries due to the emerging resistance of mosquito vectors to preventative measures. There is universal agreement that targeting the insects transmitting the infectious agent remains the mainstay in combating malaria, and novel insect growth regulators are urgently needed to supplement or replace pyrethroids and other insecticides currently used. Incisive studies of Nobo may pave the way for new agents to combat malaria by targeting the enzyme.

## Figures and Tables

**Figure 1 biomolecules-13-00976-f001:**
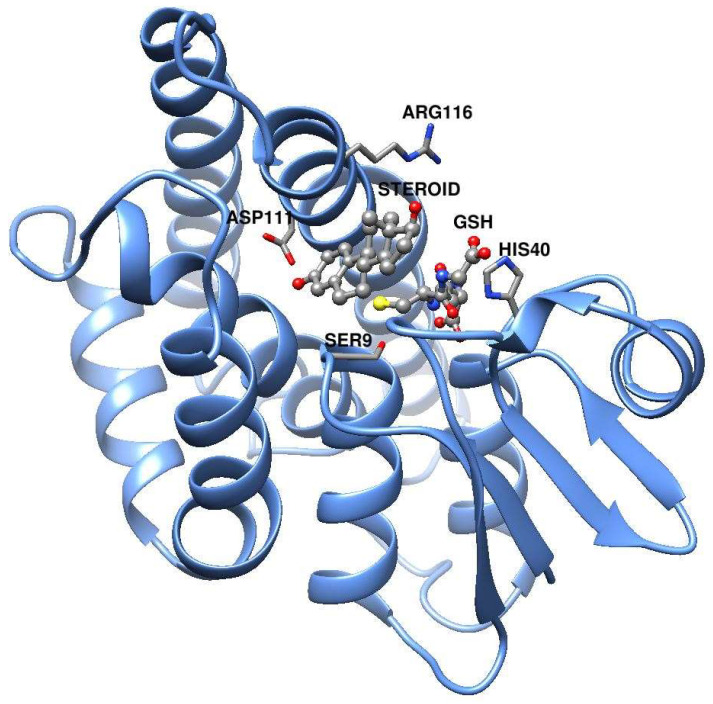
Model of *Anopheles gambiae* Nobo subunit. The structure is based on PDB ID: 6KEP, which is not shown in the picture except for the two ligands 17β-estradiol (steroid) and glutathione (GSH) rendered as ball-and-stick models and colored according to element. The steroid substrates 5-AD and 5-PD have a similar size and overall shape as 17β-estradiol, such that they presumably bind to the same site in Nobo. The functional enzyme is a structural dimer composed of two identical subunits.

**Figure 2 biomolecules-13-00976-f002:**
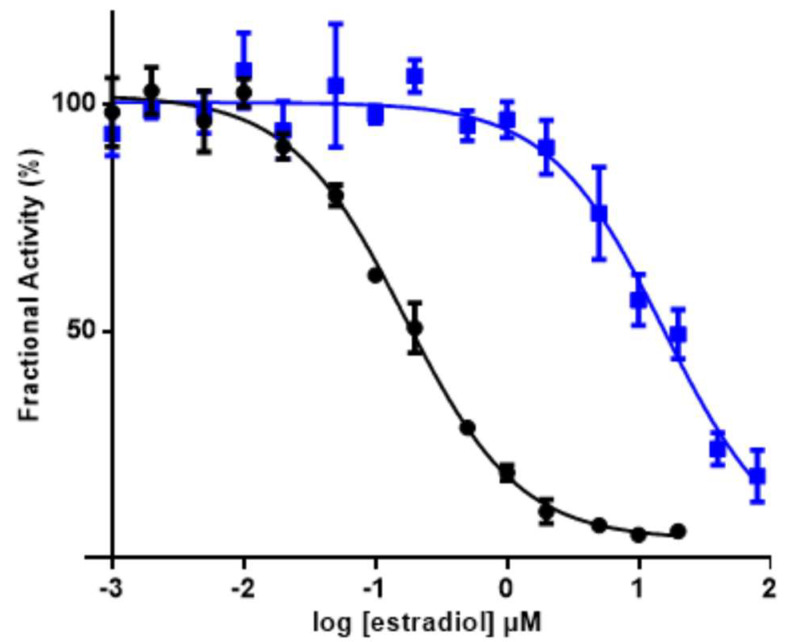
Inhibition of Nobo from *An. gambiae* with 17β−estradiol. Comparison of wild-type (black dots) with mutant Asp111Asn (blue squares) enzyme using 1 mM DCNB and 5 mM glutathione as substrates and varied 17βestradiol concentrations at pH 7.5. Measurements were carried out in triplicate at 30 °C.

**Figure 3 biomolecules-13-00976-f003:**
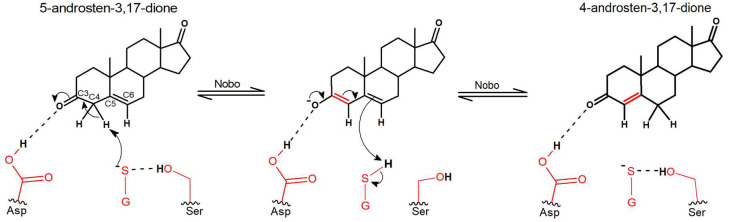
Proposed reaction mechanism of Nobo from *An. gambiae* in the double-bond isomerization of 5-androsten-3,17-dione. Asp111 in a steroid-binding pocket is polarizing the carbonyl group in the C3 position of the substrate. Ser9 is stabilizing the thiolate of glutathione (GSH), which serves as a base abstracting a proton from the C4 position.

**Figure 4 biomolecules-13-00976-f004:**
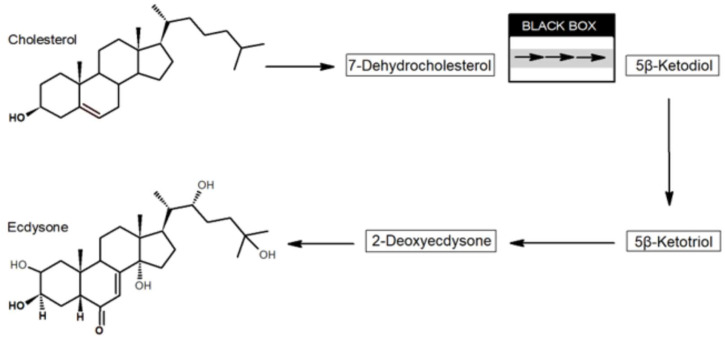
Biosynthetic pathway leading from cholesterol to the hormone ecdysone in insects. Established intermediates are indicated in boxes, whereas the black box includes several unidentified reactions and implicates the cytochrome P450 enzymes CYP6T3, CYP307A1, and CYP307A2 [31,32].

**Table 1 biomolecules-13-00976-t001:** Specific activities (μmol/min per mg) of Nobo from *An. gambiae* with alternative GST substrates. Initial reaction rates at 30 °C were determined spectrophotometrically by published procedures. The values are based on triplicate measurements.

Substrate	WT	D111N	S9A	H40N	R116A
CDNB	24.7 ± 3.4	26.2 ± 4.8	1.8 ± 0.2	14.0 ± 0.5	3.3 ± 0.1
DCNB	16.4 ± 1.9	14.7 ± 0.5	0.49 ± 0.03	5.0 ± 0.1	2.34 ± 0.02
CuOOH	5.0 ± 0.5	3.8 ± 0.4	0.05 ± 0.01	0.9 ± 0.1	0.16 ± 0.02
Allyl-ITC	3.3 ± 0.4	3.4 ± 0.5	0.2 ± 0.1	2.7 ± 0.3	2.0 ± 0.2
Phenethyl-ITC	0.4 ± 0.1	1.2 ± 0.3	0.04 ± 0.01	0.34 ± 0.02	0.1 ± 0.04
5-AD	245 ± 8	0.8 ± 0.1	16.0 ± 2.0	69.3 ± 3.7	48.1 ± 1.2
5-PD	49.7 ± 6.1	0.6 ± 0.2	5.3 ± 0.5	23.7 ± 4.1	14.7 ± 1.4

## Data Availability

Experimental data can be obtained from the authors by request.

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
