# Peer review of "Potent GST Ketosteroid Isomerase Activity Relevant to Ecdysteroidogenesis in the Malaria Vector Anopheles gambiae"

_biomolecules, 2023, doi:10.3390/biom13060976_

Round 1

Reviewer 1 Report

This is a well-written paper. Experimental aspects appear sound.

The surrounding literature is adequately discussed, placing the current work in appropriate context.

Some readers might be puzzled at the use of a human hormone in the assay of an insect hormone biosynthesis pathway. But it appears likely that an analogous reaction is catalysed by Nobo in ecdysone biosynthesis.

Suggestions:

Table 1: An. gambiae should be italicised.

A figure showing the model of Nobo and highlighting the locations of residues S9, H40, D111 and R116 would be helpful to orient the reader.

I suggest replacing...

In R116A removal of the positively charged guanidinium group of Arg is preventing the formation of an ionic bond with a carboxyl group of glutathione.

...with...

In R116A, removal of the positively charged guanidinium group of Arg prevents the formation of an ionic bond with a carboxyl group of glutathione.

Author Response

  1.  An. gambiae has been italicized in Table 1.
  2.  A figure indicating the mutated positions has been added, as suggested.
  3.  The language has been changed as suggested to: 

    In R116A, removal of the positively charged guanidinium group of Arg prevents the formation of an ionic bond with a carboxyl group of glutathione.

Reviewer 2 Report

The authors confirmed the Nobo (GST) can catalyzing double-bond isomerization in the steroids 5-androsten-3,17-dione and 5-preg-15 nen-3,20-dione, which is important for ecdysteroid biosynthesis. Further, it is confirmed that Asp111 is essential for activity with the steroids, but 18 not for conventional GST substrate. The results are interesting and with a novelty to readers, and suitable for this journal.

Concerns:

 1, Table 1, the amino acids mutations should be described clearly in the material and methods part.

 2, “Black box “ in the Biosynthetic pathway as presented in figure 3, as we known, there are many P450s (Halloween genes) that function in ecdysone biosynthesis. There should have some description or incorporated in this figure.

Author Response

  1. The mutants were, like wild-type DNA, obtained by total synthesis of the genes. The text has been modified in order to further clarify this point.
  2. Clarification of the existence of known enzymes in the Black Box has been made in the legend to the figure. 

Reviewer 3 Report

The study is very interesting and important and highlights the possibility of improving an additional method for controlling very severe human malaria whose pathogen Plasmodium falciparum can be transmited by species fo the An.gambiae Complex. 

As An. gambiae is a species Complex I need to know if Nobo's characterization was made for An.gambiae s.l. (lato sensu ) not considering the different species that make up the complex and what implications  only for Nobo characterized for An.gambiae s.l.

In the results of lines 109 to 117 the authors describe procedures that should be registered in material and methods. 

As for the form of presentation, in the results item I observe that there are discussion paragraphs. There are even records of citations in the results item. In the discussion item, few paragraphs discuss the results described in the article.  

My suggestion is to make a single item results and discussion or that in the results only these are presented and in the item discussion only  the reason for the results is described.

These are observations I make for the moment. 

Author Response

  1. We now explain that "DNA coding for Nobo protein XP_319963.1 from An. gambiae str. PEST" was used. Variant sequences of Nobo among different mosquito strains are unknown to us.
  2. With due respect for the reviewer's suggestion, we prefer to keep the lines 109 to 117 in their original place.
  3. With due respect for the reviewer's suggestion, we prefer to keep the results and discussion as originally presented.